# *SLC35A2* gene product modulates paramyxovirus fusion events during infection

**Yanling Yang**[1,2], **Yuchen Wang**[1,2], **Danielle E. Campbell**[3], **Heng-Wei Lee**[1,2], **Wandy Beatty**[1], **Leran Wang**[3], **Megan Baldridge**[3], **Carolina B. López**[1,2]*

1 Department of Molecular Microbiology, Washington University School of Medicine, St. Louis, Missouri, United States of America, 2 Center for Women Infectious Disease Research, Washington University School of Medicine, St. Louis, Missouri, United States of America, 3 Department of Medicine, Division of Infectious Diseases, Washington University School of Medicine, St. Louis, Missouri, United States of America

* clopezzalaquett@wustl.edu

**Data Availability Statement:** All data are in the manuscript and supporting information files.

**Funding:** C.L.B. was supported by NIH R01AI137062 and the BJC Investigator program at WUSTL. M.T.B. was supported by The G. Harold

## Abstract

Paramyxoviruses are significant human and animal pathogens that include mumps virus (MuV), Newcastle disease virus (NDV) and the murine parainfluenza virus Sendai (SeV). Despite their importance, few host factors implicated in paramyxovirus infection are known. Using a recombinant SeV expressing destabilized eGFP (rSeVC^dseGFP) in a loss-of-function CRISPR screen, we identified the CMP-sialic acid transporter (CST) gene *SLC35A1* and the UDP-galactose transporter (UGT) gene *SLC35A2* as essential for paramyxovirus infection. As expected, *SLC35A1* knockout (KO) cells showed drastic reduction in infections with SeV, NDV and MuV due to the lack of cell surface sialic acids receptors. However, *SLC35A2* KO cells revealed unknown critical roles for this factor in virus-cell and cell-to-cell fusion events for the different paramyxoviruses. While UGT was essential for virus-cell fusion during SeV entry to the cell, it was not required for NDV or MuV entry. Importantly, UGT promoted the formation of syncytia during MuV infection, suggesting a role in cell-to-cell virus spread. Our findings demonstrate that paramyxoviruses can bind to or enter A549 cells in the absence of canonical galactose-bound sialic-acid decorations and show that UGT facilitates paramyxovirus fusion processes involved in entry and spread.

## Author summary

Paramyxovirus entry, the first step in establishing an infection, involves two key processes: attachment and virus-cell fusion. Paramyxoviruses use their attachment protein to bind to sialic acid-containing molecules on the cell surface, which serve as receptors, followed by fusion of the viral and cellular membranes mediated by the viral Fusion protein. Few host factors involved in the attachment and, particularly, the fusion process have been identified. Using CRISPR/Cas9 screening, we identified *SLC35A2*, a gene encoding the UDP-galactose transporter, as essential for Sendai virus-mediated virus-cell fusion, but not for Newcastle disease virus or mumps virus, which are from the same family. This is the first report of a gene specifically involved in the virus-cell fusion process on paramyxoviruses. Furthermore, we found that the UDP-galactose transporter promotes mumps virus-

and Leila Y. Mathers Foundation and the Burroughs Wellcome Fund Pathogenesis of Infectious Disease Program. D.E.C. was supported by NIH T32 DK077653-29 and Crohn's & Colitis Foundation Research Fellowship Award #935619. The funders had no role in study design, data collection and analysis, decision to publish, or preparation of the manuscript.

**Competing interests:** The authors have declared that no competing interests exist.

induced cell-cell fusion, suggesting a difference in the host galactoproteins involved in virus-cell versus cell-cell fusion. Our study provides new insights into the molecular mechanisms of fusion during viral infection.

## Introduction

The Paramyxovirus family includes the major human and animal pathogens measles virus (MV), mumps virus (MuV), human parainfluenza virus (hPIV), Newcastle disease virus (NDV) and the highly pathogenic zoonotic Hendra (HeV) and Nipah (NiV) viruses, which impose a significant burden on global public health and cause substantial economic losses [1,2].

Paramyxoviruses are single-stranded negative-sense RNA enveloped viruses containing glycoproteins on their surface for attachment and fusion with the cell. These glycoproteins are essential for virus entry and infection. Attachment proteins vary across the different genera of paramyxoviruses, and can be either the glycoprotein (G), the hemagglutinin (H), or the hemagglutinin-neuraminidase (HN). The *Avulavirus* (e.g., NDV), *Rubulavirus* (e.g., MuV), and *Respirovirus* (e.g., Sendai virus (SeV)) genera attach to the cell surface through the virus HN protein that binds sialic acid-containing cell surface molecules [3]. Sialic acid-containing molecules also serve as attachment receptors for many other viruses, including influenza virus, reovirus, adenovirus, and rotavirus [4]. After the virus attaches to the host cell, the fusion (F) protein undergoes a conformational change that triggers the fusion of the host cell and viral membranes. Virus-cell membrane fusion leads to the release of the viral ribonucleoprotein complex into the cytosol, allowing for viral replication and transcription to occur [5]. In some cases, the virus enters and fuses with the endosomal membrane [6,7]. In addition, the F protein can facilitate cell-to-cell fusion and syncytia formation, for example during MuV infection [8–10].

Sialic acids are bound to carbohydrate chains on glycoproteins and glycolipids in the Golgi apparatus via different glycosidic linkages. The most common linkage types are $\alpha$2,3-linkage to a galactose residue, $\alpha$2,6-linkage to a galactose residue, $\alpha$2,6-linkage to an N-acetylgalactosamine residue, and $\alpha$2,8-linkage to another sialic acid moiety on a glycan [4]. Sialic acid and galactose are transported into the Golgi by the CMP-sialic acid transporter (CST) and the UDP-galactose transporter (UGT) encoded by *SLC35A1* and *SLC35A2*, respectively. CST facilitates the assembly of sialic acid onto glycoproteins and glycolipids [11]. SeV and MuV are reported to only use $\alpha$2,3-linked sialic acid to attach to cells [12–15], while NDV can bind to both $\alpha$2,3-linked and $\alpha$2,6-linked sialic acids [16]. All reported paramyxovirus receptors involve sialic acids linked to a galactose, suggesting that this glycan motif may be essential for sialic acid-dependent virus infection.

Targeting host factors essential to the viral lifecycle is one promising avenue for antiviral drug development [17,18]. Unfortunately, the list of known cellular host factors and their importance in modulating the paramyxovirus lifecycle is relatively sparse when compared to other common viruses such as influenza virus and coronavirus [19–22]. Even less is known about similar and divergent host protein requirements among different paramyxoviruses. Given the significance of paramyxoviruses in disease and the lack of clear candidates for a host-directed antiviral drug design, unbiased and high-throughput screening for host factor dependencies remains a necessary research objective for this virus family.

In this work, we used the murine paramyxovirus Sendai virus (SeV) which causes respiratory infection in mice and is widely used as a model paramyxovirus [23–28], to perform

CRISPR-Cas9-based screenings for essential pro-viral host factors. We leveraged a novel recombinant SeV strain expressing a destabilized eGFP (dseGFP) reporter that allowed for sensitive measurements of viral genome replication and transcription within the infected cell, pe*rmitting more accurate analysis of the CRISPR-Cas9 knockout (KO) library screening results. Consistent with several published screens for other sialic-acid dependent RNA viruses, we found that the top essential pro-viral genes included *SLC35A1* [29–32] and *SLC35A2* [32,33]. CST serves as an essential protein for the expression of the virus attachment receptors. In contrast, we discovered that UGT, in addition to contributing to virus attachment, plays independent roles in paramyxovirus virus-cell and cell-cell fusion processes.

## Results

### CRISPR knock-out screen identifies *SLC35A1* and *SLC35A2* as essential genes for Sendai virus infection

To identify host factors essential for paramyxovirus infection, we developed genome-wide CRISPR KO libraries in A549 cells to screen for infection with the model virus SeV (Fig 1A). Cas9-stable A549 cells were generated via transduction with a lentivirus expressing Cas9. Several single cell clones of A549-Cas9 cells were selected based on the expression of Cas9 as determined by western blot (S1A Fig). The Cas9 activity of the clones was then confirmed by an eGFP knockout assay where higher Cas9 activity resulted in a lower percentage of GFP positive cells (S1B Fig) [34]. The A549-Cas9 single cell clone 1 with the highest Cas9 efficiency (S1C Fig) was selected and transduced at a low multiplicity of infection (MOI) of 0.3, with the Human CRISPR KO lentiviral single guide (sg) RNA Library Brunello [35] followed by puromycin selection.

For screening, we generated a recombinant SeV expressing a destabilized(ds) eGFP protein (rSeVC$^{dseGFP}$). This virus was generated by inserting a dseGFP between SeV NP and P genes (S2A Fig). Destabilization of the protein is achieved by fusing a proline-glutamate-serine-threonine-rich (PEST) peptide to eGFP. This peptide reduces the half-life of eGFP from 20 hours to 2 hours causing a 90% signal loss [36,37]. As shown in S2B Fig, the rSeVC$^{dseGFP}$ did not show signs of attenuation in virus titer ($10^{8.28}$ vs $10^{8.35}$ TCID$_{50}$/ml) but exhibited lower eGFP intensity compared with rSeVC$^{eGFP}$ in infected A549 cells. To identify host factors regulating infection regardless of the antiviral response, we performed three screens using different immunostimulatory conditions. First, transduced cells were infected with either rSeVC$^{dseGFP}$ nonstandard viral genomes (nsVG)-negative stocks in the absence or presence of the JAK/STAT signaling inhibitor Ruxolitinib. Stocks without nsVGs lack strong immunostimulatory molecules [38], whereas drug treatment precludes interferon signaling. Second, another batch of transduced cells were infected with rSeVC$^{dseGFP}$ stock with a high content of immunostimulatory nsVGs (nsVG positive), which induce strong immune responses [39]. Among the subset of sgRNAs that were enriched in the GFP-negative cell population in all three independent screenings relative to the control (S1–S3 Tables), we identified the genes *SLC35A1* and *SLC35A2* encoding the CMP-sialic acid transporter and the UDP-galactose transporter, respectively, as significantly enriched (log fold change >2.5, p value <0.01) regardless of the antiviral response (Fig 1B–1D).

### *SLC35A1* and *SLC35A2* are essential for SeV infection

The *SLC35A1* gene encodes the CST necessary for the sialylation of proteins and lipids [11]. The *SLC35A2* gene encodes the UGT which is required not only for the galactosylation of N- and O-glycans on glycoproteins but also for the synthesis of galactosylceramide and galactosyl

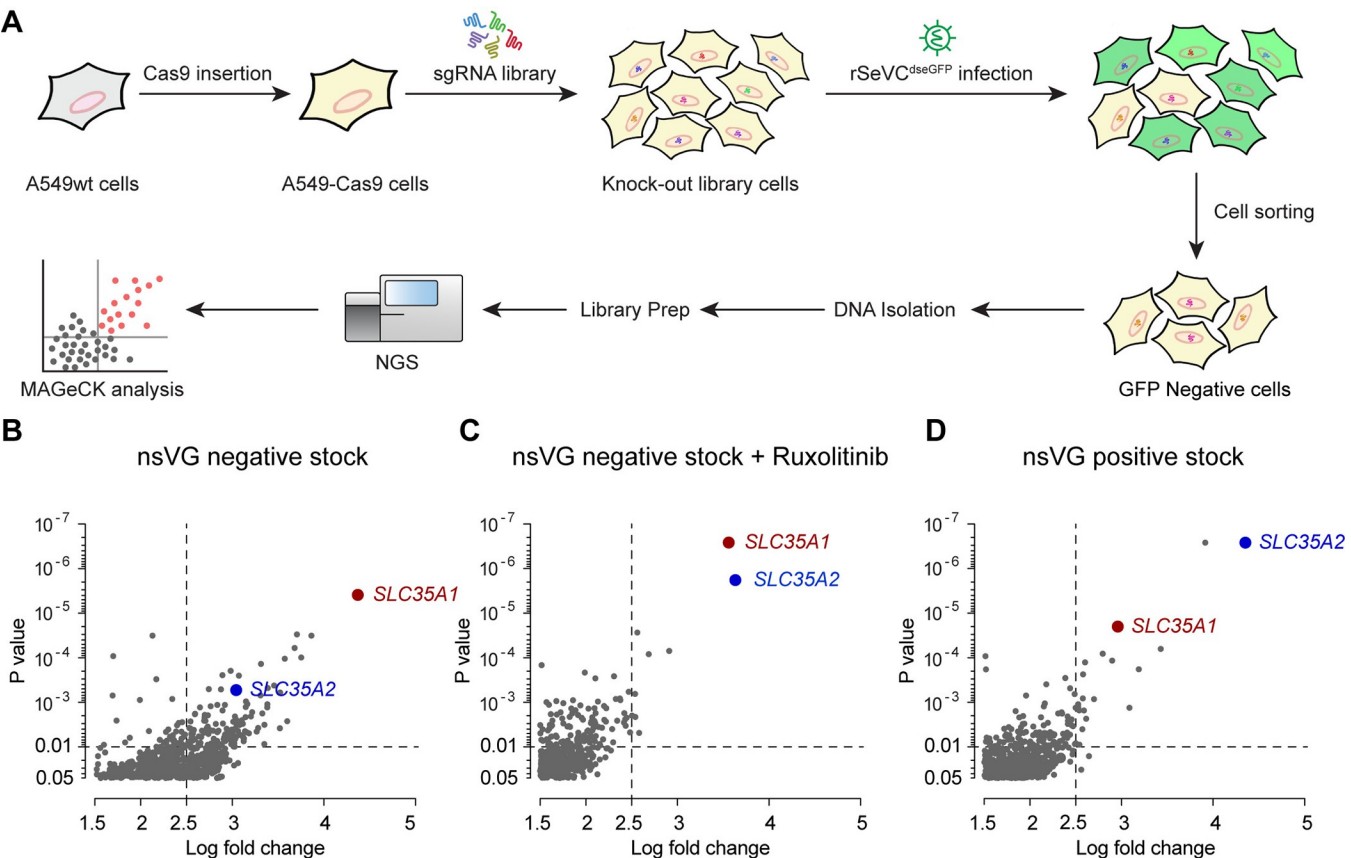

**Fig 1. CRISPR screen workflow and sgRNA enrichment analysis.** (A) Summary of the CRISPR screen workflow, from the generation of the A549-Cas9 stable cell line to the sequencing and MAGeCK analysis. (B-D) Scatter plots showing the enrichment of sgRNAs in eGFP-negative cells at 24 hpi relative to control unsorted mock cells (P value < 0.05, log2 fold change > 1.5). Cells were infected with rSeVC^dseGFP nsVG negative stock at an MOI of 10 or rSeVC^dseGFP nsVG positive stock at an MOI of 3. Differences in enrichment were calculated as log2-normalized fold change. *SLC35A1* and *SLC35A2* sgRNAs were significantly enriched in all three independent screenings: using nsVG negative virus stock (B), nsVG negative virus stock with 5uM Ruxolitinib treatment (C) and nsVG positive virus stock (D).

diglyceride [11]. CST and UGT are found in the membrane of the Golgi apparatus and transport CMP-sialic acid and UDP-galactose from the cytosol into Golgi vesicles for the generation of glycans (Fig 2A) [11]. The terminal sugar chains of sialylated glycoproteins and gangliosides that act as SeV receptors, such as GD1a and GQ1b, are shown in Fig 2B. To validate the functional significance of *SLC35A1* and *SLC35A2* during SeV infection, these genes were individually knocked out in A549-Cas9 cells using sgRNAs specific for each gene. A control cell line was made by transducing a scramble sgRNA that did not target any specific host gene. Transduced cells were then selected with puromycin followed by single cell cloning. To validate the knockouts, we tested for the presence of surface sialic acids and galactose in the KO cell lines by staining with the lectins Sambucus nigra agglutinin (SNA) and Erythrina cristagalli lectin (ECL) to detect cell-surface sialic acid and galactose, respectively as previously described [31,40] (Fig 2C). As shown in Fig 2D, *SLC35A1* KO cells lack cell surface sialic acid while having more exposed galactose [40]. *SLC35A2* KO cells lack galactose in the cell surface and as expected since most of terminal sialic acid are linked to galactoses, have a significantly reduced level of sialic acid. In addition, to assess the impact of knocking out the transporter in the whole cell glycome, cellular glycans in the KO cell lines were evaluated using a lectin array assay. As shown in S3 Fig, reduced levels of total cellular sialic acid were detected in both KO

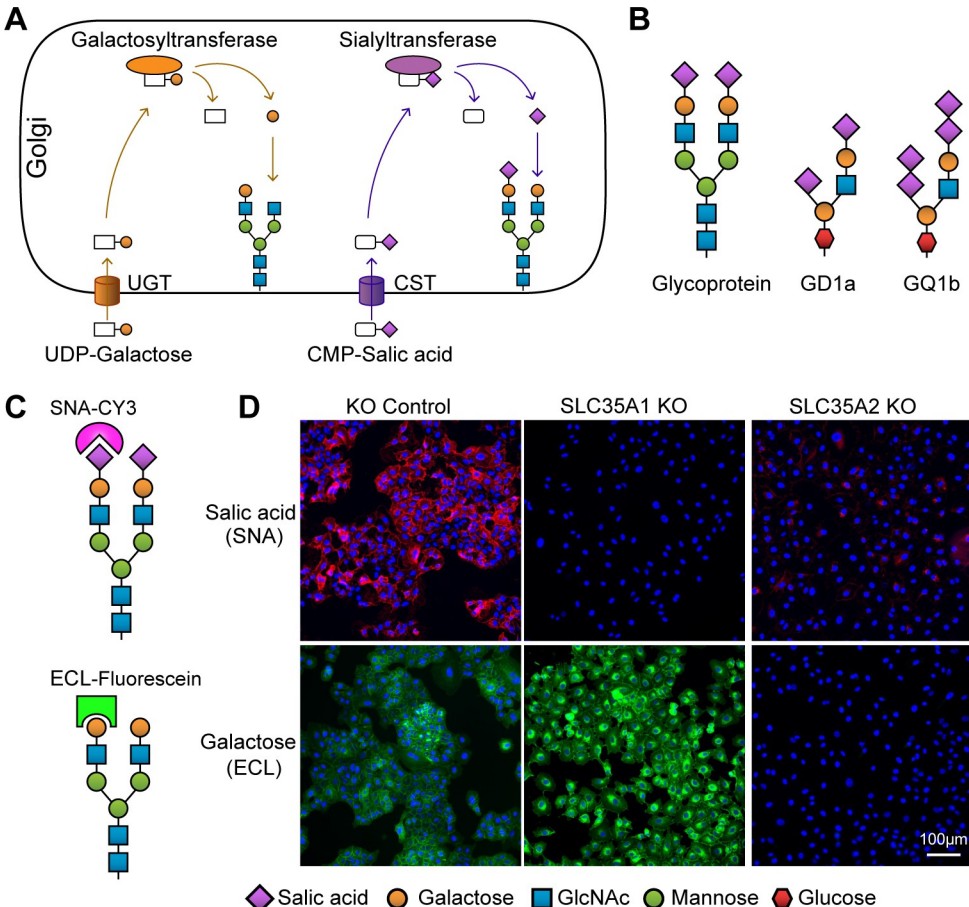

**Fig 2. Lectin staining in A549 *SLC35A1* and *SLC35A2* KO cells.** (A) Schematic of glycosylation pathways in the Golgi apparatus, showing the roles of the *SLC35A1* gene encoding CMP-sialic acid transporter (CST) and the *SLC35A2* gene encoding UDP-galactose transporter (UGT). Galactosyltransferase and sialyltransferase enzymes add galactose and sialic acid residues to glycans, respectively. (B) Examples of SeV receptors: glycoprotein and gangliosides GD1a and GQ1b, illustrating the incorporation of sialic acid and galactose residues. (C) Diagram of lectin staining: SNA (Sambucus nigra agglutinin) binds to cell surface sialic acid, while ECL (Erythrina cristagalli lectin) binds to galactose. (D) Analysis of sialic acid and galactose expression by lectin staining. A549 control, *SLC35A1* KO, or *SLC35A2* KO cells were fixed and stained with fluorophore-conjugated lectins SNA or ECL specific for sialic acid (magenta) or galactose (green) and analyzed by widefield microscopy. Scale bar lengths are indicated. Images are representative of three independent experiments.

cell lines by the lectins SAMB and SNA, agreeing with our staining results. Similarly, reduced binding of Maackia amurensis lectin (MAA/MAL) demonstrates decreased levels of total cellular galactose. Other glycans, including high-mannose-containing glycans detected by 3 lectins: the Burkholderia cenocepacia lectin (BC2L-A), Griffithia sp. lectin (GRFT), and Oryza sative lectin (ORYSATA), were also identified as altered in whole cell extracts. The role of mannose-containing glycans in paramyxovirus remains to be determined.

We then used a SeV reporter virus expressing eGFP (rSeVC^eGFP) to directly assess the impact of *SLC35A1* or *SLC35A2* during infection. As a control, we used vesicular stomatitis virus (VSV) which does not depend on sialic acid for entry [31,41]. GFP expression at 24 hpi with an MOI of 1.5 for SeV and an MOI of 0.015 for VSV was used as a readout of infection and virus replication. Absence of *SLC35A1* and *SLC35A2* resulted in loss of SeV infectivity in most cells, with only a few cells showing viral replication (Fig 3A). We confirmed absence of

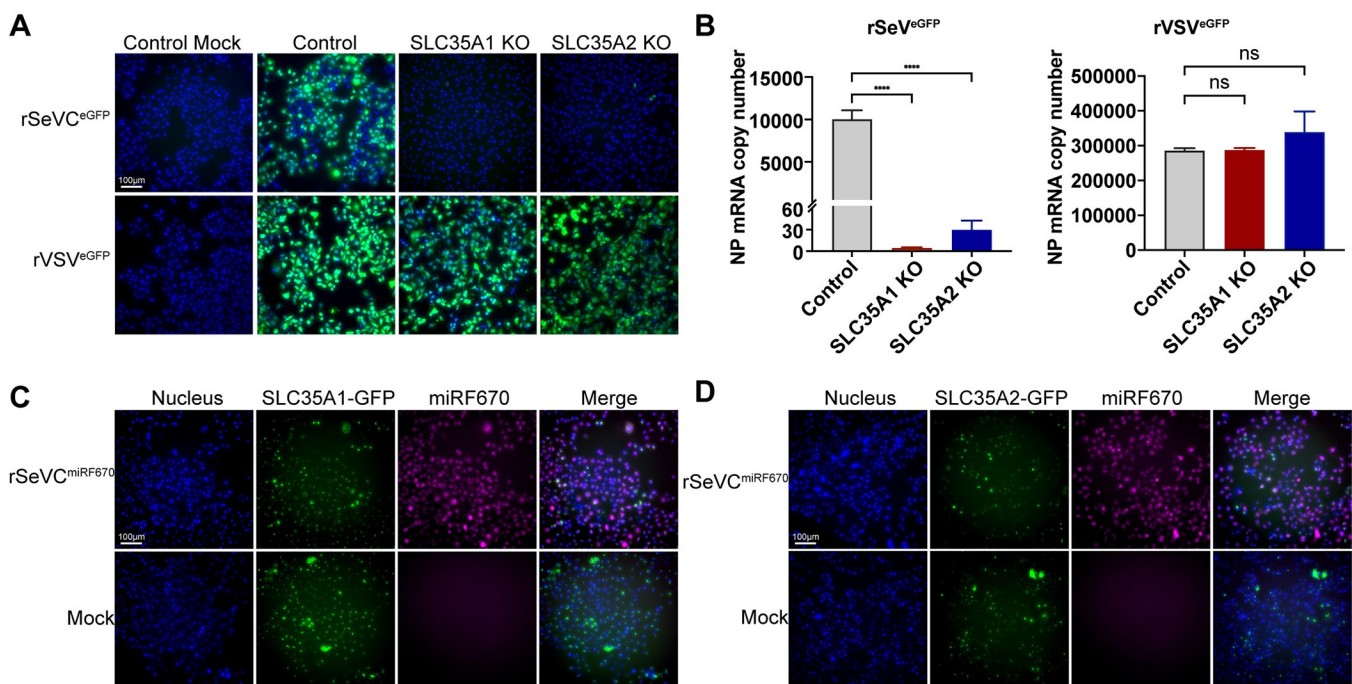

**Fig 3.** ***SLC35A1* and *SLC35A2* are essential for optimal SeV infection.** Fluorescence images showing GFP expression in A549 control, *SLC35A1* KO, and *SLC35A2* KO cells 24hpi with rSeVC$^{eGFP}$ at an MOI of 1.5 or rVSV$^{eGFP}$ at an MOI of 0.015. Scale bar lengths are indicated. (B) Quantification of viral NP mRNA copy numbers by relative qPCR analysis in control and KO cells infected with rSeVC$^{eGFP}$ at MOI 1.5 or rVSV$^{eGFP}$ at MOI 0.015 and harvested at 24hpi. Expression of mRNA was calculated relative to the housekeeping index with *GAPDH* and *β-actin*. Data represent the mean of three independent experiments. ****: p<0.0001, ns: not significant. (C, D) Widefield images showing complementation of KO cells with either (C) *SLC35A1*-GFP or (D) *SLC35A2*-GFP infected with rSeVC$^{miRF670}$ (magenta) at MOI of 3 and harvested at 24hpi. The nucleus was stained with Hoechst 33342 (Blue). Scale bar lengths are indicated. Images are representative of three independent experiments.

SeV replication in *SLC35A1* KO cells and drastically reduced replication in *SLC35A2* KO cells by evaluating SeV NP mRNA expression by qPCR (Fig 3B). In contrast, VSV infection proceeded normally in the absence of *SLC35A1* or *SLC35A2*. To exclude the possibility of other defects that may result in the restriction of SeV infection in the KO cells, we complemented *SLC35A1* KOs with cDNA expressing *SLC35A1*-GFP and *SLC35A2* KOs with cDNA expressing *SLC35A2*-GFP. Since the complemented *SLC35A1* and *SLC35A2* were fused with GFP, in these experiments we used a SeV miRF670 reporter virus rSeVC$^{miRF670}$ in which a monomeric near-infrared fluorescent protein (miRFP670) was inserted into the SeV genome between the NP and P genes. At 24 hpi with rSeVC$^{miRF670}$, we observed recovered SeV replication in complemented KOs (Fig 3C and 3D). Infection using high MOIs of rSeVC$^{dseGFP}$ or rSeVC$^{eGFP}$ showed more cells infected in *SLC35A1* KO cells, and an even larger number in *SLC35A2* KO cells at 24hpi, but in both cases the percentage of cells infected was significantly less than controls, suggesting that SeV can enter more cells independent of *SLC35A1* and *SLC35A2* when used at high MOIs (S4 Fig). Overall, these data confirmed the critical, yet not completely overlapping, roles of *SLC35A1* and *SLC35A2* during SeV infection.

## *SLC35A2* differentially impacts infection with diverse paramyxoviruses

To assess whether the observed non-overlapping functions of *SLC35A1* and *SLC35A2* were maintained during infection with other paramyxoviruses, we infected single and double KO A549 cells with the *Respirovirus* rSeVC$^{eGFP}$, the *Avulavirus* rNDV$^{eGFP}$, or the *Rubulavirus* MuV at an MOI of 1.5 and looked for either eGFP (SeV and NDV) or MuV NP expression at

24 hpi. The double KO cells were made by transducing A549 *SLC35A1* KO cells with *SLC35A2* sgRNA and it was confirmed by lectin staining (Fig 4A). In all cases, the absence of *SLC35A1* or both *SLC35A1* and *SLC35A2* resulted in drastically reduced viral infectivity, with only a few cells showing expression of the viral reporter gene. Interestingly, for both NDV and MuV infections, but not SeV, more infected cells were observed in *SLC35A2* KO cells than in double KO cells (Fig 4B and 4C), suggesting that *SLC35A2* and *SLC35A1* have non-redundant functions during paramyxovirus infection and that *SLC35A2* differentially impacts infection with different paramyxoviruses.

## *SLC35A2* differentially impacts SeV, NDV, and MuV infection and spread in A549 cells

To further investigate the impact of *SCL35A2* on paramyxovirus infection and spread, we followed the infection in *SCL35A2* KO cells through a 4-day infection period (Fig 5). Interestingly, infections in *SLC35A2* KO cells displayed varied phenotypes across these viruses. As shown before, SeV infection was drastically reduced to one or two cells per image (5X magnification) and the virus did not spread throughout the time course (Fig 5A and 5B). NDV could infect a larger proportion of *SLC35A2* KO cells before the infected cells died (S5 Fig), but there was no evidence of virus spread, compared to NDV replication in control cells (Fig 5C and 5D). In contrast, MuV infected and spread well in *SLC35A2* KO cells as evidenced by staining for the virus NP (Fig 5E and 5F). Taken together, these data indicate that *SLC35A2* plays differential roles in the infection and spread of different paramyxoviruses during infection in A549 cells.

## *SLC35A2* is not essential for virus attachment or viral genome replication during SeV infection

We next focused on investigating where *SLC35A2* played a critical role during the virus infection cycle. Based on the organization of the terminal sugar chains of sialylated glycoproteins and the apparent requirement for galactose for sialic acid proximal binding (Fig 2B), we began by evaluating whether *SLC35A2* impacts the attachment of SeV to the cell surface. As expected, *SLC35A1* KO cells exhibited robust restriction of SeV binding evidenced by consistently negative cell surface HN staining in infected group after co-incubation of virus and cells for 1 hour at 4°C. However, incubation of SeV with *SLC35A2* KO cells under the same conditions resulted in positive cell surface HN staining, albeit lower than control cells (Fig 6A and 6B), suggesting that the impact of *SLC35A2* on paramyxovirus infection extends beyond the virus binding step.

To confirm that *SLC35A2* did not directly affect SeV genome replication and transcription, we took advantage of the recombinant reporter SeV virus rSeVC$^{eGFP\Delta FHN+GFtail}$. This virus was made by removing the original SeV F and HN and inserting a chimeric VSV glycoprotein G fused with the C-terminal tail of SeV F (GFtail) thus replicating as SeV but entering the cells as VSV (S6A Fig). As expected, rSeVC$^{eGFP\Delta FHN+GFtail}$ can infect *SLC35A1* KO cells using the VSV GFtail protein for entry (S6B Fig). Then we asked whether rSeVC$^{eGFP\Delta FHN+GFtail}$ can replicate without *SLC35A2*. Although both viruses infect control cells to a similar degree, rSeVC$^{eGFP}$ was unable to infect and replicate in *SLC35A2* KO cells but rSeVC$^{eGFP\Delta FHN+GFtail}$ infected and spread normally in these cells (Fig 6C). In addition, SeV transcription measured by SeV NP mRNA expression showed that absence of *SLC35A2* does not impair the viral polymerase activity (Fig 6D). These data demonstrate that *SLC35A2* increases the efficiency of SeV binding but is not essential for virus attachment or viral genome replication and transcription.

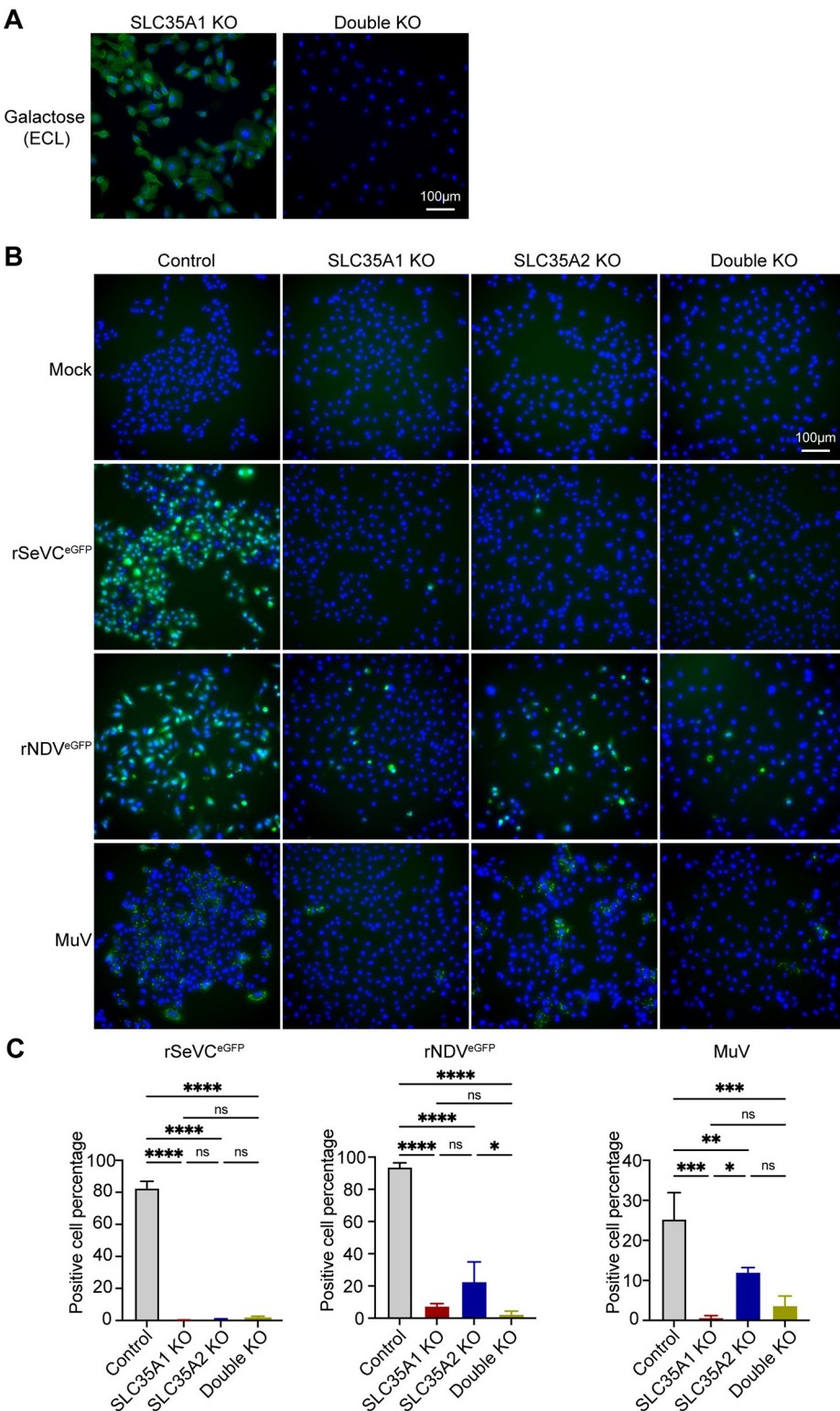

**Fig 4. Impact of *SLC35A1* and *SLC35A2* single KOs or double KOs in infection with different paramyxoviruses.**
(A) Analysis of galactose expression in A549 *SLC35A1/SLC35A2* double KO cells by lectin staining. *SLC35A1* KO cells, and double KO cells were fixed and stained with ECL followed by fluorescence microscopy. The images show the distribution of galactose (green) residues. The nucleus was stained with Hoechst 33342 (Blue). Scale bar lengths are indicated. Images are representative of three independent experiments. (B) Control and KO cell lines were infected

with rSeVC^eGFP, rNDV^eGFP, or MuV at an MOI of 1.5. Widefield fluorescence images showing GFP expression (rSeVC^eGFP and rNDV^eGFP) or NP staining (MuV) in control, *SLC35A1* KO, *SLC35A2* KO, and double KO cells at 24hpi. Scale bar lengths are indicated. Images are representative of three independent experiments. (C) Quantification of infected cells in (B). Statistics were calculated with an ordinary one-way ANOVA. ns: not significant, *: p<0.05, **: p<0.01, ***: p<0.001, ****: p<0.0001.

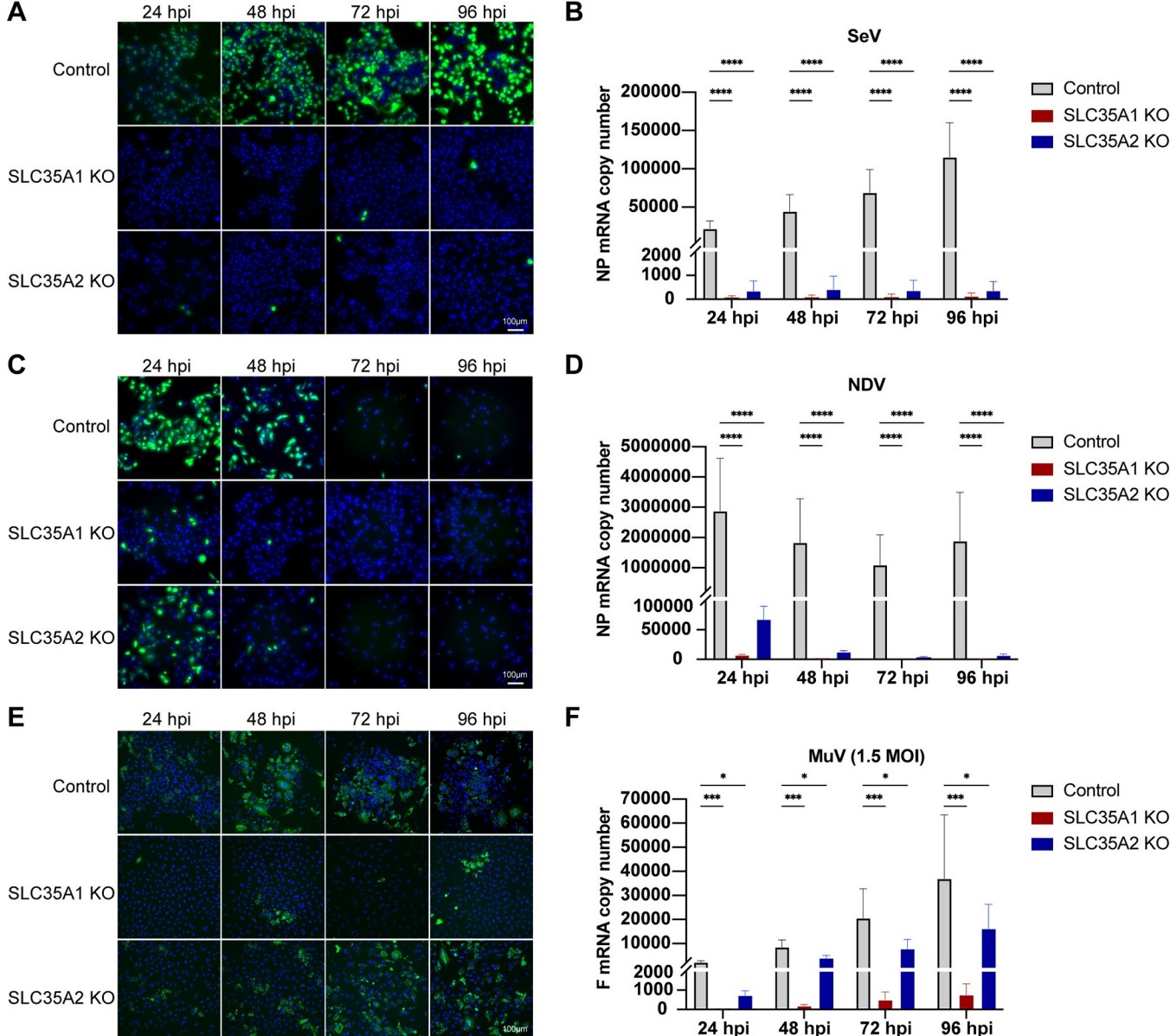

**Fig 5. Time course infection with paramyxoviruses in *SLC35A1* and *SLC35A2* KO A549 cells.** (A and C) Fluorescence or (E) immunofluorescence images of A549 control, *SLC35A1* KO, and *SLC35A2* KO cells infected with (A) rSeVC^eGFP, (C) rNDV^eGFP, or (E) MuV at an MOI of 1.5. Images were analyzed at 24, 48, 72, and 96 hpi. The nucleus was stained with Hoechst 33342 (Blue), green fluorescence indicates viral infection represented by eGFP or MuV NP staining. Images are representative of three independent experiments. (B, D, and F) Expression of viral NP or F mRNA relative to the housekeeping index from cellular RNA collected 24–96 hpi for each indicated virus. Data represent the mean of three independent experiments. Statistics were calculated with a two-way ANOVA. ns: not significant, *: p<0.05, ***: p<0.001, ****: p<0.0001.

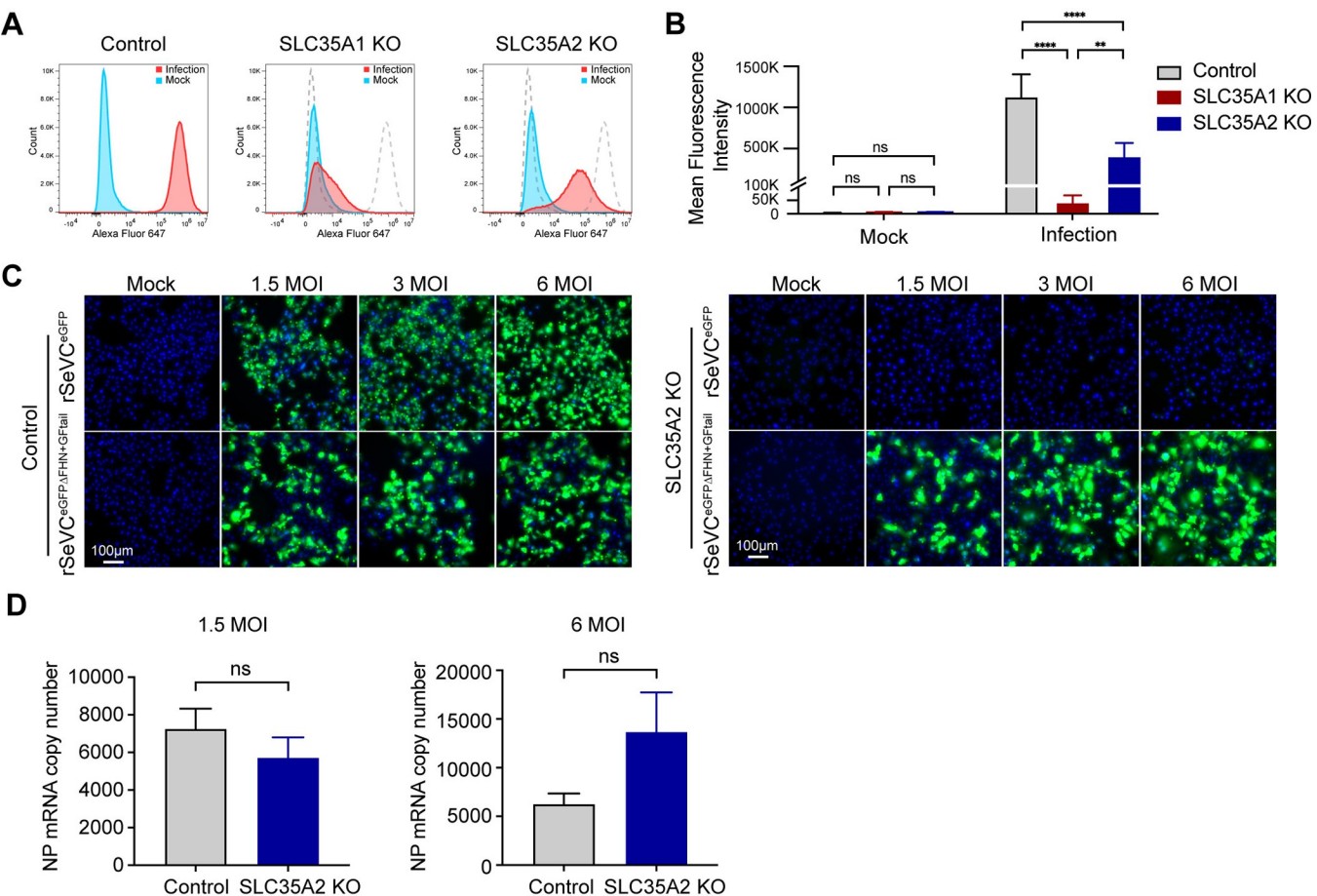

**Fig 6. *SLC35A2* is not essential for SeV attachment or genome replication.** (A) Flow cytometry analysis of SeV binding in A549 control and KO cells. Cells were either mock-infected (blue) or infected with Sendai virus (red) and stained with anti-HN-Alexa 647 antibody. The dashed lines in the middle and right panels represent the histograms of the control cell line. Data shown are representative of four independent experiments. (B) Mean fluorescence intensity of (A). Statistics were calculated with a two-way ANOVA. ns: not significant, **: $p<0.01$, ****: $p<0.0001$. (C) A549 control or *SLC35A2* KOs cells infected with rSeVC$^{eGFP}$ or rSeVC$^{eGFP\Delta FHN+GFtail}$ for 24hpi at MOIs of 1.5, 3, or 6. The nucleus was stained with Hoechst 33342 (blue). Green fluorescence indicates reporter gene expression. Scale bar lengths are indicated. Images are representative of three independent experiments. (D) qPCR analysis of A549 control cells or *SLC35A2* KOs infected with rSeVC$^{eGFP\Delta FHN+GFtail}$ for 24 hpi at MOIs of 1.5 or 6. Expression of SeV NP mRNA was calculated relative to the housekeeping index. Data represent the mean of three independent experiments. Statistics were calculated with unpaired t-test. ns: not significant.

## *SLC35A2* is essential for virus-cell fusion during SeV infection

To directly test the impact of *SLC35A2* in the SeV virus-cell fusion process, we tested for intracellular detection of the SeV internal M (matrix) protein as a marker for fusion [42] using a recombinant SeV rSeV-M-HA [43] in which the M protein is fused with an HA tag (Fig 7A). In brief, SeV was incubated with cells at a high MOI at 37°C for 3 hours to allow virus entry and fusion. The cells were then treated with Proteinase K followed by fixation, permeabilization, and staining of SeV M and envelope HN proteins. Proteinase K treatment was used to remove cell surface-attached viral particles and the HN protein staining was used to visualized virus attached to the cell surface. As shown in Fig 7B, SeV HN was detected in both cell lines prior to proteinase K treatment, similar to Fig 6A, but not after proteinase K treatment. However, the M protein was detected in the control cells but not in *SLC35A2* cells, regardless of proteinase K treatment, suggesting that *SLC35A2* is essential for efficient fusion of the virus and cell membranes during virus entry.

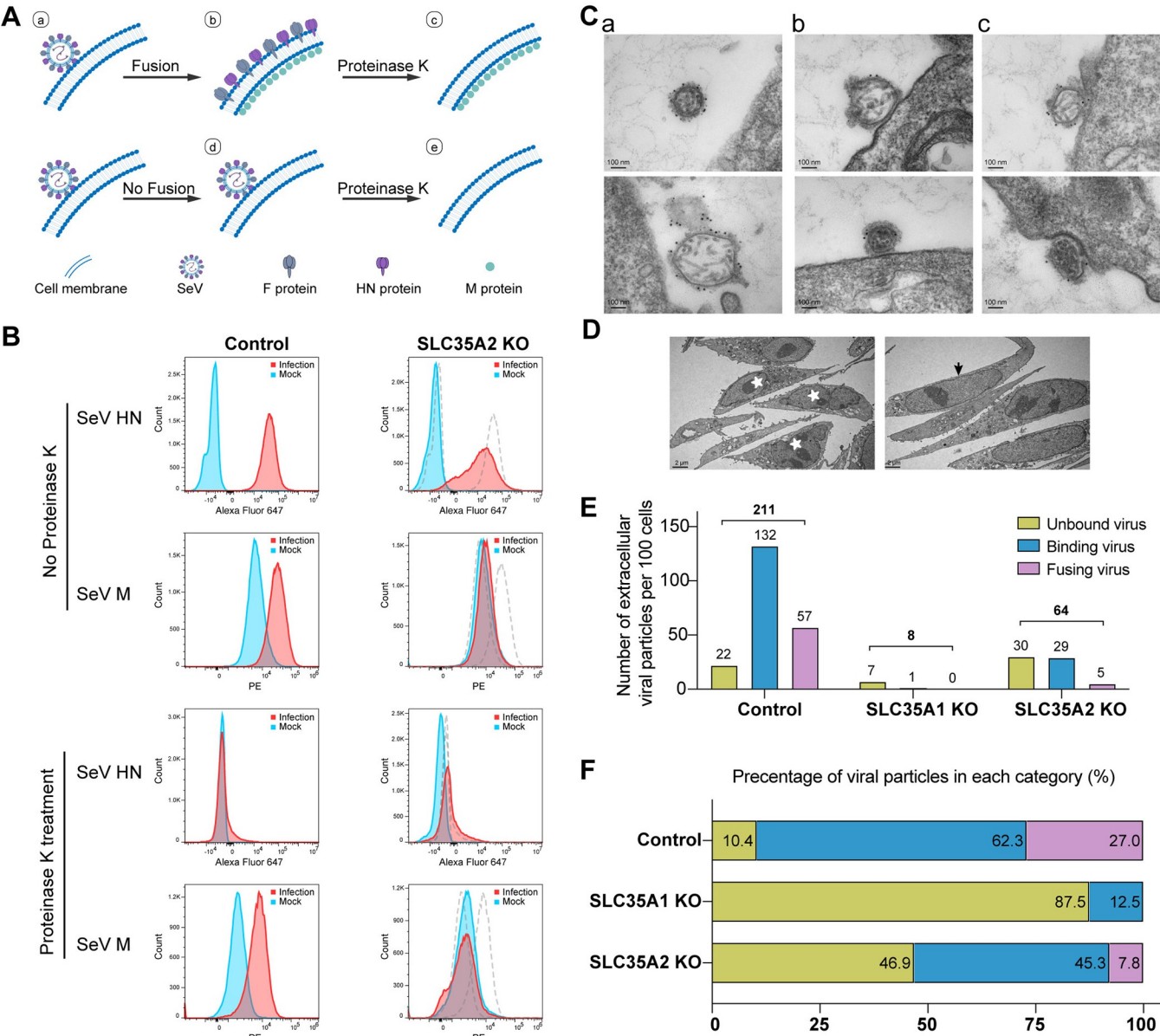

**Fig 7. *SLC35A2* is essential for SeV-cell fusion.** (A) Illustration of the principle behind the M protein detection fusion assay. a: SeV attaches to the cell membrane. b. The virus fuses with the cell membrane, with F and HN proteins distributed on the cell surface, and the M protein located on the inner side of the membrane. c.Proteinase K treatment destroys surface F and HN proteins, while the M protein remains inside the cell and can be detected by flow cytometry. d. Proteinase K treatment also releases cell surface-bound but unfused viruses that are then washed away. e. No intracellular M protein is detected in cells where SeV does not fuse. (B) Flow cytometry histograms of A549 control or *SLC35A2* KOs stained with anti-HA and anti-HN after incubation with rSeV-M-HA at an MOI of 400 (infection) or mock infection (Mock) at 37˚C for 3 hours, followed by 70 minutes of proteinase K treatment (0.5mg/ml). The dashed lines in the right panels represent the histograms of the control cell line. Data shown represent one of three independent experiments. (C-F) Electron microscopy images of A549 control, *SLC35A1* KO, or *SLC35A2* KO cells labeled with anti-F and anti-HN antibodies followed by anti-IgG conjugated to gold beads after incubation with rSeVC^dseGFP at an MOI of 800 at 4˚C for 1 hour. (C) Viruses were classified by their distance to cells and categorized into three classes: a, Unbound virus; b, Binding virus: no space between the virus and the cell, with a visible gap between the double-layer membranes of both; c, Fusing virus: no space between the virus and the cell, with the double-layer membranes of the virus and cell indistinguishable at any point of contact. Scale bar lengths are indicated. (D) Examples of analyzed cells cross-sections through the nucleus are marked by white stars, while excluded cells are indicated by black arrows. Images were captured at direct magnifications of 60,000X (C) and 2,000X (D). Scale bar lengths are indicated. (E) Quantification of the number of virus particles observed in 100 cell cross-sections per sample. The number of total virus particles and of each category of virus are indicated above the bars. (F) Percentage distribution of cells with different virus categories in control, *SLC35A1* KO, and *SLC35A2* KO cells.

To directly visualize the fusion process between SeV and the cell membrane using electron microscopy (EM), we incubated A549 control and KO cells with SeV at an MOI of 800 to ensure viruses could be found on cross-sections. Viruses labeled with SeV F and HN antibodies were categorized into three classes based on their distance from the cells, as illustrated in Fig 7C: unbound virus, binding virus, and fusing virus. For analysis, 100 cells with cross-sections through the nucleus, indicating the cell center, were randomly selected for each sample, as marked by white stars in Fig 7D. Cells close to one another, indicated by black arrows, were excluded. Out of the 100 randomly selected cells sections in each sample, 42, 32, and 7 cell cross-sections had virus associations in control, *SLC35A1* KO, and *SLC35A2* KO cells, respectively. As shown in Fig 7E, a total of 211, 8, and 64 viruses were detected in the Control, *SLC35A1* KO, and *SLC35A2* KO samples, respectively. Only 8 viruses were found in *SLC35A1* KO cells, consistent with the binding results that reflect a lack of the sialic acid receptor. In *SLC35A2* KO cells, binding viruses were fewer than in control cells, which aligns with Figs 6A and 6B and 7B, showing that *SLC35A2* KO decreased virus-cell binding efficiency. Fusing viruses were also reduced in *SLC35A2* KO cells, and the binding-to-fusion ratio in *SLC35A2* KO cells was 5/29, representing 40% of the ratio in control cells (57/132). The percentages of each virus category are summarized in Fig 7F. These findings, together with the fusion results detected by intracellular M protein, demonstrate that *SLC35A2* plays a critical role in facilitating virus-cell fusion during SeV infection in A549 cells.

## *SLC35A2* is implicated in syncytia formation during MuV infection

As MuV infected and transcribed in *SLC35A2* KO cells (Fig 5E and 5F), we next asked whether *SLC35A2* impacted other steps of the MuV replication cycle such as infectious virion production and syncytia formation. As shown in Fig 8A and 8B, there was no significant difference in the virus production of infectious viral particles between control and *SLC35A2* KO cells infected at MOI of 1.5 or 15, indicating that *SLC35A2* does not impact MuV infectious particle production. Interestingly, we noticed less syncytia formation in *SLC35A2* KO cells at both low and high MOI. To confirm this difference, we quantified the number of syncytia formed under low and high MOI conditions at 48 hpi. We defined an NP-positive large cell with more than three nuclei as syncytia. As shown in Fig 8C–8E, *SLC35A2* KO cells formed less syncytia compared to control cells at both MOIs, suggesting that *SLC35A2* is implicated in MuV-induced syncytia formation suggesting a role in cell-to-cell virus spread.

## Discussion

This manuscript reports the identification of *SLC35A1* and *SLC35A2* as non-reductant proviral host genes during paramyxovirus infection. *SLC35A1* is essential for the attachment of SeV, NDV, and MuV to the cell due to its role in exposing the viral receptor sialic acid on the cell surface, similar to what has been described for influenza virus and porcine delta coronavirus [29,31]. Although the role of *SLC35A2* is assumed to be related to virus attachment given that galactose is typically considered the sugar to which sialic acid is linked [4], we found that *SLC35A2* is not essential for virus attachment but it improves virus-cell binding efficiency. However, *SLC35A2* is crucial for the fusion of SeV with the target cell and for MuV induced cell-to-cell fusion and syncytia formation, suggesting a specific role for this protein in fusion events during virus infection.

Virus-cell fusion is an important target for antiviral drug development. To date, research on the fusion process and anti-fusion strategies have mainly focused on the role the viral proteins F and HN play during viral infection [44,45]. However, our understanding of the host factors involved in the processes of virus-cell and cell-to-cell fusion that occur during infection

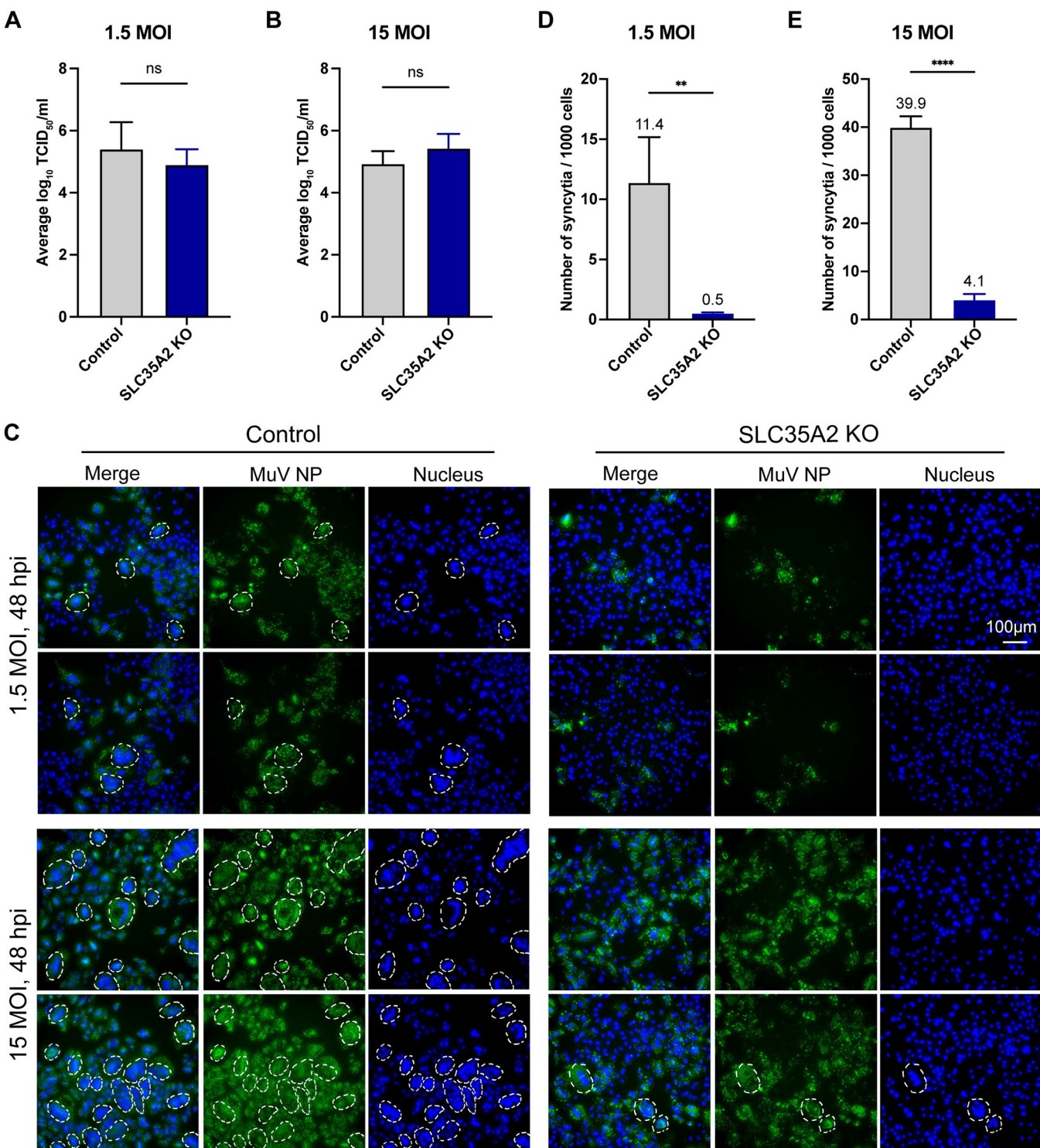

**Fig 8. Measurement of MuV infectious particle production and MuV-induced syncytia.** (B) MuV tissue culture infectious dose 50 ($TCID_{50}$) in supernatants of A549 control and *SLC35A2* KO cells infected for 72 h at an MOI of 1.5 (A) or 15 (B). Data represent the mean of three independent experiments. Statistics was calculated by unpaired t-tests. ns: not significant. (C) Syncytia formation 48hpi in A549 control and *SLC35A2* KO cells infected with MuV at MOIs of 1.5 and 15. The white dashed lines mark syncytia, which was defined as MuV NP positive cells containing more than 3 nuclei. Images are representative of three independent experiments. (D and E) Number of syncytia among 1000 cells 48hpi at MOIs of 1.5 and 15. Data represent the mean of three independent experiments. Statistics were calculated by unpaired t-test. **: $p<0.01$, ****: $p<0.0001$.

are limited to the described role of 25-hydroxycholesterol in interfering with NiV induced cell-to-cell fusion [46], and the role of soluble N-ethylmaleimide-sensitive factor attachment protein receptor (SNARE) protein USE1 in the glycosylation and expression of MuV fusion protein [47]. To our knowledge, no host factors have been identified to affect paramyxovirus virus-to-cell fusion.

The paramyxoviruses used in this study failed to infect *SLC35A1* KO cells due to the lack of cell surface sialic acid receptors. Interestingly, although SeV failed to infect *SLC35A2* KO, it attached to these cells but less efficiently than to control cell. The ability of SeV to attach to cells lacking galactose suggests that it utilizes alternative sialylated surface receptors beyond the classic sialic acid linked to galactose [4,12,13,31,48–53]. Alternative linkages, such as α2,6-linkage to N-acetylgalactosamine have been reported [4,54]. SeV and MuV are reported to only use α2,3-linked sialic acid to attach to cells [12–15], while NDV can bind both α2,3-linked and α2,6-linked sialic acids [16]. Based on the data reported here, we hypothesize that either α2,6-linkage to N-acetylgalactosamine or other sialic acid linked molecules contribute to SeV entry into A549 cells.

The role of *SLC35A2* in RNA virus infection remains unclear. A study showed that absence of *SLC35A2* abolishes influenza H1N1 replication [33] and another study showed no negative effect on influenza H7N9 virus binding and internalization and instead showed decreased influenza virus polymerase activity using a viral replicon system [32]. In addition, *SLC35A2* has been identified as an HIV X4 strain-specific restriction factor in primary target CD4+ T cells [55]. Our SeV and MuV genome replication data suggests *SLC35A2* has no effect on viral replication and transcription during infection with these viruses. As we have confirmed that *SLC35A2* affects paramyxovirus virus-cell and cell-to-cell fusion, there is a possibility that this gene also affects fusion processes in infections with influenza and other viruses.

Specific N-glycans are important for the fusion activity of the F protein of several paramyxoviruses [56–58]. In this study, the virus stocks used to infect cells were infectious, even when MuV particles were produced in *SLC35A2* KO cells, suggesting that during paramyxovirus infection virus-cell fusion and cell-cell fusion event are influenced by changes in host cells. The specific molecular mechanisms involved require further investigation. A whole cell lectin array assay corroborated a reduction in galactose-containing glycans in the *SLC35A2* KO cells, as shown by reduced binding Maackia amurensis lectin (MAA/MAL). However, differently from results of lectin staining of the cell surface, ECA/ECL bound similarly to extracts from control cells and *SLC35A2* KO cells likely reflecting differences between cell surface glycans and total cell glycans. In addition, the whole cell lectin array revealed that other cellular glycans are impacted in this cell line. For example, binding of Solanum tuberosum lectin/agglutinin (STA/STL), which binds oligomers of *N*-acetylglucosamine, was reduced in *SLC35A2* KO cells. Of note, lectin array data on *SLC35A2* knockdown cells have demonstrated that glycan changes are cell type-dependent [59]. Since only A549 cells were used in this study, further investigation is needed to determine whether the observed impact of *SLC35A2* KO on virus-cell and cell-cell fusion are cell type-dependent.

During MuV infection, syncytia formation occurs late in the virus life cycle when the viral glycoproteins expressed on infected cells mediate fusion with neighboring cells [60]. Viral components can be transmitted between the fused infected cells through syncytia. We observed that despite lower MuV F mRNA at late times post-infection in *SLC35A2* KO cells compared to controls, the amount of infectious viral particles produced in both cell lines at 48 hpi was the same, suggesting an important role for cell-to-cell fusion in MuV spread. This finding suggests that *SLC35A2* promotes MuV cell-to-cell transmission through MuV induced syncytia formation. Interestingly, *SLC35A2* deficiency didn't seem to impair MuV-cell fusion but significantly decreased MuV induced cell-cell fusion, suggesting differences between the

mechanisms of virus-cell fusion and cell-cell fusion. The ability of viruses to trigger cell-cell fusion between infected cells and surrounding non-infected target cells is mainly related to the expression of the viral F at the surface of the infected cells. Cell-cell fusion is then mediated by interactions of viral F with surface molecules or receptors expressed on neighboring non-infected cells. A two-component system, in which the first component consists of transmembrane receptors and the second component consists of the multiprotein fusion complexes, can mediate cell-cell fusion events [61]. Cell surface fusion proteins such as fertilin and meltrin α may involve in this process [62]. *SLC35A2* KO may influence the glycosylation of host glycoproteins required for SeV-cell fusion and MuV-induced cell-cell fusion, but not for MuV-cell fusion. Therefore, further investigation into the specific host-side mechanisms is needed.

In conclusion, our CRISPR-Cas9 KO screen identified *SLC35A1* and *SLC35A2* as genes encoding critical host proteins for paramyxovirus infection. *SLC35A1* is essential for viral binding, while *SLC35A2* is crucial for SeV-cell fusion and is implicated in MuV-induced cell-to-cell fusion. Our findings show that even without *SLC35A2*, SeV can bind, NDV can enter and express genes, and MuV can complete its life cycle, indicating that galactose is not essential for viral attachment. This suggests the existence of sialic acid-bound receptors not yet characterized. In addition, the reduced syncytia formation in *SLC35A2* knockout cells at low MOI suggests that *SLC35A2* may be important for MuV cell-to-cell transmission. These insights highlight the potential of targeting *SLC35A2* for therapeutic interventions against paramyxovirus infections.

## Materials & methods

### Cell lines

A549 cells (ATCC, #CCL-185), BSR-T7/5 cells (kindly provided by Dr. Conzelmann) [63], Lenti-293T cells (TaKaRa, #632180), and LLC-MK2 cells (ATCC, #CCL-7) were cultured in tissue culture medium (Dubelcco's modified Eagle's medium (DMEM) (Invitrogen, #11965092) supplemented with 10% fetal bovine serum (FBS) (Sigma, #F0926), gentamicin 50 ng/ml (ThermoFisher, #15750060), L-glutamine 2 mM (Invitrogen, #G7513) and sodium pyruvate 1 mM (Invitrogen, #25-000-C1) at 5% $CO_2$ 37˚C. Cells were treated with mycoplasma removal agent (MP Biomedical, #3050044) and tested monthly for mycoplasma contamination using the MycoAlert Plus mycoplasma testing kit (Lonza, #LT07-318).

### Genetically modified cell lines

Lentiviruses for transduction were generated by co-transfecting a plasmid expressing the gene of interest or sgRNA together with the lentivirus packaging plasmids psPAX2 (Addgene, #12260) and pMD2.G (Addgene, #12259) into Lenti-293T cells using a TransIT-Lenti Transfection Reagent (Mirus Bio, #MIR 6604). Supernatants were collected 48 hours post transfection and lentivirus were detected by a Lenti-X GoStix Plusx kit (TaKaRa, #631280). A549 cells were then transduced with 500ul supernatants containing the lentivirus and 8 ug/ml polybrene (1200rpm, 30˚C, 2 hours). The cells were transferred to 6 well plates the second day followed by antibiotic selection (Blasticidin 10 ug/ml for 1 week, Puromycin 0.5 ug/ml for 1 week, Hygromycin 400ug/ml for 2 weeks). Surviving cells were single cell cloned and confirmed by western blot or lectin staining.

A549-Cas9 stable line were generated by transducing lentiCas9-Blast plasmid (Addgene, #52962) [64] into A549 wt cells followed by Blasticidin selection and single cell cloning. A549-*SLC35A1*, A549-*SLC35A2* KO, and A549 control cell lines were made by transduction of sgRNA plasmids (Puromycin) to A549-Ca9 stable cell line. sgRNA plasmids with Puromycin resistance for *SLC35A1* (sgRNA: CCATAGCTTTAAGATACACA), *SLC35A2* (sgRNA:

TGCGGGCGTAGCGGATGCTG) or a scrambled sgRNA control (CACTCACATCGCTA-CATCA) were ordered from Applied Biological Materials Inc. The *SLC35A1* and *SLC35A2* double KO cell line was made by transduction of a *SLC35A2* sgRNA plasmid (Hygromycin) to A549-*SLC35A1* KO cells.

## Viruses and virus infection

SeV Cantell expressing the reporter miRF670 (rSeVC[miRF670] [65]), SeV expressing HA-tagged M (rSeV-HA-M [43]) and the NDV reporter virus rNDV[eGFP] were grown in 10-day-old, embryonated chicken eggs (Charles River) for 40 hours as previously described [66]. The VSV reporter virus (rVSV-eGFP) [67,68] was obtained from Dr. Sean Whelan (Washington University in St.Louis) and grown in BSR-T7/5 cells. MuV (ATCC, #VR-1379) was grown in LLC-MK2 cells.

Infections with SeV, NDV, or MuV were performed after washing the cells once with PBS and incubating with virus diluted in infection media (DMEM, 35% bovine serum albumin (Sigma, #A7979), penicillin-streptomycin (Gibco, #15140–122), and 5% NaHCO$_3$ (Gibco, #25080094) at 37°C for 1 hour, shaking every 15 minutes. Cells were then washed twice with PBS and supplemented with additional infection media. The infected cells were incubated at 37°C until harvest.

## Recombinant reporter viruses rescue

The reporter viruses rSeVC[eGFP] and rSeVC[dseGFP] were rescued using the SeV Cantell strain reverse genetic system as described before [65]. First, two full-length plasmids pSL1180-rSeV--C[eGFP] and pSL1180-rSeV-C[dseGFP] were made by replacing the miRF670 gene of pSL1180-rSeV-C[miRF670] with an eGFP or a destabilized eGFP (dseGFP) gene [36]. Additional nucleotides were inserted downstream of the dseGFP gene to ensure that the entire genome followed the "rule of six". The viruses were rescued by co-transfecting full-length plasmids and the three helper plasmids to BSR-T7/5 cells using Lipofectamine LTX with Plus Reagent (Invitrogen, #15338100). The expression of GFP or dseGFP was monitored daily using fluorescence microscopy. At 4 days post-transfection, the cell cultures were harvested, and the supernatants were used to infect 10-day-old specific-pathogen-free embryonated chicken eggs via the allantoic cavity after repeated freeze-thaw cycles. After incubation for 40 hours at 37°C, 40–70% humidity, the allantoic fluids were harvested and the TCID$_{50}$ was measured using LLC-MK2 cells.

The reporter virus rSeVC[GFP-ΔFHN+GFtail] virus was generated and rescued by replacing SeV F and HN gene with a VSV G gene while retaining the SeV F protein tail. In brief, a VSV-GFtail plasmid was made by replacing G tail (G protein 490-511aa) with SeV F tail (F protein 524-565aa) using plasmid pMD2.G (Addgene, #12259), then the pSL1180-rSeVC[GFP-ΔFHN+GFtail] full-length plasmid was made by cloning GFtail and deleting SeV F and HN gene from pSL1180-rSeVC[eGFP] through PCR and In-Fusion cloning (TaKaRA Bio, #638948). Virus was rescued as described above, and after five serial passages on A549-*SLC35A1* KO cells, virus titer increased from 10$^2$ to above 10$^7$ TCID$_{50}$/mL. Sanger sequencing confirmed the rescued virus sequence but revealed a D99G mutation within the M protein.

## Cas9 expression assay

Protein was extracted from A549-Cas9 stable cell line pool and single cell clones using 1% NP40 lysis buffer as described previously [65]. After a 20-minute incubation on ice and high-speed centrifugation for 20 minutes at 4°C, supernatant was collected, and protein concentration was quantified using the Pierce BCA Protein Assay Kit following the user's guidelines

(Thermo Fisher, #23225). Next, 30 ug protein was denatured for 5 minutes at 95˚C, loaded in a 4% to 12% Bis Tris gel (Bio-Rad, #3450124), and transferred to a PVDF membrane (Millipore Sigma, #IPVH00010). After blocking with 5% milk, membranes were incubated overnight with anti-CRISPR-Cas9 (Abcam, #ab191468) or anti-GAPDH (Sigma, #G8795) antibodies diluted in 5% BSA containing TBS with 0.1% Tween20. Membranes were incubated with anti-mouse secondary antibody conjugated with HRP for 1 hour in 5% BSA in TBST. Membranes were developed using Lumi-light western blotting substrate (Roche, #12015200001) and HRP was detected by a ChemiDoc (Bio-Rad).

## Cas9 activity assay

A549-Cas9 single cell clones were transduced with pXPR_011 lentivirus at an MOI of ~1.0 in 12 well plates. pXPR_011 plasmid was a gift from John Doench & David Root [34] (Addgene, #59702). Transduced cells were transferred to 6 well plates on day 3 post transduction and treated with puromycin. On day 9 post transduction, surviving cells from each single cell clone were collected and eGFP signal was detected by spectral flow cytometry. Active Cas9-expressing lines resulted in a reduction of eGFP when transduced with pXPR_011 as this vector delivers both eGFP and a sgRNA targeting eGFP. Because eGFP is linked to puromycin gene with a 2A site, abrogation of eGFP will have no impact on puromycin resistance. The lower eGFP percentage of a single cell clone indicates higher Cas9 activity.

## A549-Brunello CRISPR KO library

A549-Cas9 stable cells were transduced at a low MOI (~0.3) with Human CRISPR Knockout Pooled Library Brunello (Addgene, #73178) [35]. Transduction conditions and antibiotic concentration were optimized for the A549-Cas9 stable cell line (cell seeding density: $8 \times 10^4$ per well (6-well plate); Puromycin concentration: 0.5ug/ml. Lentivirus library was tittered to achieve a 30–50% infection rate, and transductions were performed with $1.35 \times 10^8$ cells to achieve a representation of at least 500 cells per sgRNA per replicate. Puromycin was added 2 days post transduction and was maintained for 5–7 days. The library cells containing sgRNA were used for CRISPR screening. Throughout the screen, the cells numbers were maintained at over $4 \times 10^7$ cells to ensure coverage of at least 500 cells per sgRNA.

## CRISPR screening

A549-Brunello library was infected by SeV reporter virus rSeVC^dseGFP nsVG negative stock at an MOI of 10 or rSeVC^dseGFP nsVG positive stock at a MOI of 3 and cells were harvested followed by cell sorting to isolate the eGFP negative cell population. After screening, the sorted cells and two aliquots of the original CRISPR library (Mock) were pelleted and frozen at -80˚C. Genomic DNA (gDNA) was isolated using the QIAamp DNA Blood Midi (Qiagen, #51183) or QIAamp DNA Bloop Mini (Qiagen, #51104) kit according to the manufacturer's instructions. The concentration of each gDNA was measured on a Qubit Fluorimeter with the Qubit dsDNA Quantification Assay Kit (Invitrogen, #Q32851). CRISPR sgRNA barcodes were amplified by PCR of up to 10 μg of each gDNA template with a P5 stagger forward primer and P7 barcoded reverse primer [69]. In addition to gDNA, each 100 μL reaction contained 4 μL Titanium Taq DNA Polymerase (Takara, #639242), 10 μL Titanium PCR buffer, dNTPs at a final concentration of 100 μM, 5 μL DMSO (Sigma, #D9170-5VL), and P5 forward and P7 reverse primers each at a final concentration of 1 μM. PCRs were run for 5 minutes at 95˚C, followed by 29 cycles of 95˚C for 30 s, 59˚C for 30 s, and 72˚C for 20 s, and a final extension step at 72˚C for 10 minutes. PCR amplicon concentrations were quantified on a Qubit Fluorimeter with the Qubit dsDNA Quantification Assay Kit before pooling and purifying with

**Table 1. CRISPR screening read counts per sample.**

| Sample | Read count |
| --- | --- |
| **nsVG negative stock** | 21,778,516 |
| **nsVG negative stock + Ruxolitinib** | 22,690,825 |
| **nsVG positive stock** | 22,583,165 |
| **Mock** | 60,290,626 |

Agencourt AMPure XP SPRI beads (Beckman Coulter, #A63880) according to the manufacturer's instructions. The final pool was submitted to the DNA Sequencing Innovation Lab (Washington University School of Medicine) for sequencing on the Illumina NextSeq-Mid platform with a 15% spike-in of PhiX DNA, yielding a total of 109,525,309 reads (Table 1). After demultiplexing according to the barcode sequences, the genes enrichment between cell populations was analyzed by MAGeCK (Version 0.5.9). The output tables were loaded and visualized with Prism 10.

## Lectin staining

Cells were seeded at a confluency of $1 \times 10^5$ cells/well in a 12-well plate a day prior to staining. The next day, the cells were washed twice with PBS and fixed using 2% PFA at RT for 15 minutes. Following fixation, the cells were blocked with 3% BSA (in PBS) at RT for 1 hour. Lectin SNA-CY3 (VectorLabs, #CL-1303-1) or ECL-Fluorescein (VectorLabs, #FL-1141-5) was diluted in PBS at a 1:500 dilution and incubated with the cells on ice for 1 hour. The nuclei were then stained with a 1:10,000 dilution of Hoechst 33342 (Invitrogen, #H3570) at RT for 10 minutes.

## Lectin array

Lectin array assays were performed using the Lectin Array 70 kit (RayBiotech, #GA-Lectin-70) according to the manufacturer's instructions. In brief, $1 \times 10^6$ cells from each cell line were collected and washed twice with PBS. The cells were then lysed in 300 μL of ELISA lysis buffer (RayBiotech, #NC0996662), and the protein concentration was determined using the Pierce BCA Protein Assay Kit (Thermo, #23225). Next, 100 μL of cell lysate (containing 1–2 mg/mL of total protein) from each cell line was dialyzed in PBS (pH 8.0) at 4°C for 3 hours, with two buffer changes. After dialysis, protein concentration was measured. A total of 30 μg of protein was labeled with biotin and dialyzed. The glass slide was dried and blocked before 100 μL of 10-fold diluted biotin-labeled samples were added. The slides were incubated at 4°C overnight. After incubation, the slides were washed and then incubated with Cy3-equivalent dye-streptavidin according to the kit protocol. Visualization was performed using an Axon GenePix 4000B scanner. Data extraction was carried out using the GAL file provided with the kit. For signal intensity normalization, control cells were defined as the "reference," and data from other arrays were normalized based on positive control wells according to the protocol.

## RNA extraction and RT-qPCR

Total RNA of infected cells and control samples were extracted using Kingfisher and a Mag-MAX mirVana Total RNA Isolation Kit (Thermo Fisher, #A27828) following manufacturer's guidelines. 300-500ng of total RNA was used for cDNA synthesis with high-capacity RNA to cDNA kit (Thermo Fisher, #18080051). qPCR was performed using SYBR green (Thermo Fisher, #S7564) and 5 μM of reverse and forward primers (Table 2) for SeV NP, NDV NP, MuV F, and VSV NP genes on an Applied Biosystems QuantStudio 5 machine. Primers used

**Table 2. Primers for qPCR.**

| Gene | Forward (5'-3') | Reverse (5'-3') | Reference |
|------|-----------------|-----------------|-----------|
| SeV NP | TGCCCTGGAAGATGAGTTAG | GCCTGTTGGTTTGTGGTAAG | [71] |
| NDV NP | CAACAATAGGAGTGGAGTGTCTGA | CAGGGTATCGGTGATGTCTTCT | [72] |
| MuV F | TCTCACCCATAGCAGGGAGTTATAT | GTTAGACTTCGACAGTTTGCAACAA | [73] |
| VSV NP | ATGACAAATGGTTGCCTTTGTATCTACTT | ACGACCTTCTGGCACAAGAGGT | [74] |

for qPCR are listed in Table 2. Relative copy numbers were normalized to human *GAPDH* and human *β-actin* expression as described previously [70].

## *SLC35A1* and *SLC35A2* complementation

To complement *SLC35A1* or *SlC35A1* to KO cells, KO cell lines were transduced with the *SLC35A1*-GFP or *SLC35A2*-GFP cDNA expression constructs. In brief, *SLC35A1*-GFP and *SLC35A2*-GFP were obtained from plasmids pEGFP.N3-*SLC35A1*-GFP and pEGFP.N3-*SLC35A2*-GFP (the plasmids were a gift from Somshuvra Mukhopadhyay Addgene plasmid # 186281 ; Addgene plasmid # 186284)[75]. We performed codon optimization for the sgRNA binding sites of the target genes and switched to pLenti-Hygro plasmid backbone pLenti CMV Hygro DEST (Addgene, 17454) for lentivirus packaging. Then, *SLC35A1*-GFP and *SLC35A2*-GFP were separately introduced into their KO cell lines using the lentiviral transduction system. Following hygromycin selection, cell lines expressing the complemented genes were obtained, namely A549-*SLC35A1* KO+A1-GFP and A549-*SLC35A2* KO+A2-GFP. Subsequently, we infected these two cell lines with SeV reporter virus rSeVC$^{miRF670}$ at a MOI of 3 and observed viral replication using fluorescence microscopy.

## NDV induced cell death quantification

To quantitatively analyze NDV-induced cell death, cells infected with rNDV-eGFP were collected at 24 and 48 hpi and analyzed by Cytek flow cytometry. In brief, debris and dead cells floating in the supernatant were collected by spin down. The attached cells were collected after trypsinization. Then cells from each condition were merged and stained with eBioscience Fixable Viability Dye eFluor 506 (Thermo Fisher, #65-0866-14) in 1:400 dilution on ice for 10 minutes. Cells were fixed with 2% PFA for 10 minutes at room temperature followed by flow analysis.

## MuV immunofluorescence

The infected cells were fixed using 2% PFA at RT for 15 minutes at specific time points post-infection followed by permeabilizing with 0.2% Triton X-100 (Sigma-Aldrich, #X100) for 10 minutes. Anti-MuV NP antibody (Thermo Fisher, #6008) was diluted in PBS at a 1:500 dilution and incubated at RT for 1 hour. Secondary antibody was diluted in PBS at a 1:500 dilution and incubated at RT for 30 minutes. The nuclei were stained with a 1:100,000 dilution of Hoechst 33342 (Invitrogen, #H-3570) along with the secondary antibody.

## Sendai virus binding experiment

To determine the binding capability of the Sendai virus to KO cells, we performed a virus-cell binding assay. Briefly, KO cells or control cells were incubated with the virus at a MOI of 30 at 4°C for one hour. The cells were then fixed with 4% paraformaldehyde (PFA) (Fisher Scientific, #50-980-495) for 10 minutes, blocked with 3% BSA for 30 minutes, and stained with a

HN Monoclonal Antibody-Alexa Fluor 647 (Thermo, #51-6494-82) for 30 minutes. Results were analyzed using Cytek flow cytometry. During the procedure, cells were washed in PBS supplemented with 2% BSA and 2 mM EDTA (Corning, #46-034-CI) 3 times between each step.

## Sendai virus Fusion experiment by flow cytometry

To detect SeV fusion with the cell membrane, we followed a published procedure to determine Ebola fusion [42]. In brief, control cells or KO cells were incubated with 400 MOI rSeV-M-HA at 37˚C for 3 hours, then treated with 0.5mg/ml proteinase K (NEB, #P8107S) at 37˚C for 70 minutes to remove virus bound to the cell surface but not fused with the cells. Next, the samples were fixed and permeabilized with eBioscience Foxp3 / Transcription Factor Staining Buffer Set (Invitrogen, #00552300). The intracellular M-HA protein was then detected by a HA-PE mAb (Biolegend, #901518). SeV HN was detected using a HN Monoclonal Antibody-Alexa Fluor 647 (Thermo, #51-6494-82) as a control for cell surface proteins. Results were analyzed by spectral flow cytometry (Cytek). Detection of M protein in the proteinase K-treated group indicated that the virus particles have fused with the cells.

## Sendai virus Fusion detection by electron microscopy

Electron microscopy was used to directly visualize the fusion process between SeV and the cell membrane. A549 control cells, *SLC35A1* cells and *SLC35A2* KO cells were seeded onto culture plates and allowed to reach approximately 70–80% confluency before infection. rSeVC$^{\mathrm{dseGFP}}$ was added to the cells at a MOI of 800 and incubated at 4˚C for 1 hour to allow viral attachment. Afterward, the cells were washed twice with PBS to remove the unbound viruses and shifted to 37˚C for 45 minutes to promote virus-cell fusion. Then the cells were fixed with 4% PFA at room temperature for 1 hour. For immunolabeling to identify SeV, monolayers were washed with PBS, blocked with 5% FBS/5% NGS for 30 minutes and subsequently incubated with primary antibodies (mouse anti-SeV F clone 11H12 at 1:2000 (provided by Thomas Moran), humanized anti-SeV hIgG1 F clone 11H12 at 1:1000, and humanized anti-SeV HN hIgG1 clone 1A6 at 1:1000 (provided by Andrew Duty at Icahn School of Medicine at Mount Sinai) for 1 hour. Following washes in block buffer, samples were incubated with the appropriate secondary antibodies goat anti-mouse IgG antibody conjugated to 6 nm colloidal gold and goat anti-human IgG antibody conjugated to 12 nm colloidal gold (Jackson ImmunoResearch Laboratories, #115-195-166 and #109-205-088) for 1 hour. Monolayers were washed and fixed in 2.5% glutaraldehyde (Ted Pella Inc., #18420) in 100 mM cacodylate buffer, pH 7.2 for 1 hour. Monolayers were gently scraped and centrifuged into an agarose pellet. For ultrastructural analyses, samples were postfixed in 1% osmium tetroxide (Ted Pella Inc. #18459)/ 1.5% potassium ferricyanide (Sigma, #13746-66-2) for 1 hour. Samples were then rinsed extensively in dH$_2$0 prior to en bloc staining with 1% aqueous uranyl acetate (Ted Pella Inc.#19481) for 1 hour. Following several rinses in dH$_2$0, samples were dehydrated in a graded series of ethanol and embedded in Eponate 12 resin (Ted Pella Inc. #75-56-9). Ultrathin serial sections of 95 nm were cut with a Leica Ultracut UCT ultramicrotome (Leica Microsystems Inc., Bannockburn, IL), stained with uranyl acetate and lead citrate, and viewed on a JEOL 1200 EX transmission electron microscope (JEOL USA Inc., Peabody, MA) equipped with an AMT 8-megapixel digital camera and AMT Image Capture Engine V602 software (Advanced Microscopy Techniques, Woburn, MA).

## MuV infectious particles measurement by TCID$_{50}$

A549 cells were seeded at 20,000 per well in a 96-well plate the day before the experiment. The collected samples were serially diluted 10-fold in infection media. The plated cells were washed

once with PBS, and 100 μL of each dilution was added to the cells, with each sample tested in triplicate. After incubating in a 37˚C incubator for 4 days, the cells were stained using the immunofluorescence method described above. The $TCID_{50}$/ml was calculated based on the fluorescence results.

## MuV-induced syncytia quantification

After immunofluorescence staining, 3 images of each sample were captured using an inverted fluorescence microscope at both 20x and 5x magnifications. The 5x images were used for quantifying the MuV-induced syncytia. Fiji software was used to count the total number of nuclei in each image, and the number of syncytia was manually marked and counted. Syncytia were defined as MuV-NP positive giant cells containing more than three nuclei. The number of syncytia per 1000 cells was then calculated. The results included three biological replicates.

## Statistics

Statistics were calculated using GraphPad Prism Version 10 (GraphPad Software, San Diego, CA). All data for the graphs can be found in S4 Table.

## Supporting information

**S1 Fig. A549-Cas9 stable cell line.** (A) The Cas9 expression of 6 A549-Cas9 single cell clones, A549wt cells, and A549-Cas9 pool was detected by western blot. GAPDH expression was determined as loading control. (B) Diagram of Cas9 activity assay: with Cas9, GFP, and sgRNA targeting GFP in the same cells, higher Cas9 activity leads to lower GFP intensity, and lower Cas9 activity leads to higher GFP intensity. (C) Percentage of GFP-positive cells detected by flow cytometry after transduction of the pXPR_011 plasmid into A549 wt cells and A549-Cas9 single cell clones. A549 wt cells without transduction were used as a negative control. The red dashed line indicates 30% of GFP positive cells; it is generally accepted that Cas9 cells with less than 30% GFP-positive cells can be used for CRISPR screening.
(TIF)

**S2 Fig. SeV reporter viruses rSeV^eGFP and rSeV^dseGFP.** (A) Schematic representation of the SeV reporter viruses. eGFP or dseGFFP gene was inserted into the SeV genome between NP and P gene. dseGFP is made by fusion of a PEST degradation sequence to the C terminal of eGFP. (B) Fluorescence images of A549wt cells infected with rSeV^eGFP and rSeV^dseGFP. A549wt cells were infected with an MOI of 3 of rSeV^eGFP or rSeV^dseGFP and images were analyzed at 24hpi. The nucleus was stained with Hoechst 33342 (Blue), green fluorescence indicates viral infection eGFP or dseGFP expression and accumulation. The images display three different fields of view. Scale bar lengths are indicated.
(TIF)

**S3 Fig. Cellular glycans analysis by lectin array.** (A) Lectin array images of control and KO cell lines. Cell lysates were analyzed using a Lectin-70 array, with selected sialic acid- or galactose-binding lectins highlighted. (B) Log2 fold change heat map. A heat map showing lectin array intensities for the three different cell lines. Fluorescence intensities were extracted from the lectin arrays using the Gal.file. Data for *SLC35A1* KO and *SLC35A2* KO cells were normalized to the control using positive control wells in each array. Log2 fold change was calculated based on normalized fluorescence intensities relative to control cells. Red indicates increased binding, while green indicates decreased binding.
(TIF)

**S4 Fig. *SLC35A1* or *SLC35A2* KO cells infected with SeV at high MOI.** Fluorescence images showing GFP expression in control, *SLC35A1* KO, and *SLC35A2* KO cells infected with rSeV-C$^{eGFP}$ or rSeVC$^{dseGFP}$ at MOIs of 5, 20, or 100, 24hpi. Scale bar lengths are indicated. (TIF)

**S5 Fig. Analysis of cell death induced by NDV infection.** (A) Flow cytometry analysis of cell debris and cells in control, *SLC35A1* KO and *SLC35A2* KO cells infected with rNDV$^{eGFP}$ at an MOI of 1.5 and control mock cells at 24 and 48 hpi. The percentages of cell debris or cells are indicated within each plot. Data shown represent one of three independent experiments. (B) Quantification of cell debris percentages at 24 and 48 hpi. Data represent the mean of three independent experiments. Statistical significance is indicated as follows: ****$p < 0.0001$, ns = not significant. (C) Flow cytometry analysis of cell death of cells population from (A) in control, *SLC35A1* KO, *SLC35A2* KO, and mock-infected cells at 24 and 48 hpi, as indicated by Fixable Viability Dye eFluor 506 staining. Percentages of dead cells are indicated within each plot. Data shown represent one of three independent experiments. (D) Quantification of dead cells percentages at 24 and 48 hpi. Data represent the mean of three independent experiments. Statistical significance is indicated as follows: *$p < 0.05$, ns = not significant. (TIF)

**S6 Fig. rSeVC$^{eGFP\Delta FHN+GFtail}$ rescue and confirmation.** (A) Schematic representation of the rSeVC$^{eGFP\Delta FHN+GFtail}$. rSeVC$^{eGFP\Delta FHN+GFtail}$ was designed by replacing SeV F and HN gene with a GSV-G deleting its tail and fusing with SeV F tail on rSeVC$^{eGFP}$. Red inverted triangle indicated a mutation on M protein. (B) Fluorescence microscopy images of *SLC35A1* KO cells infected with rSeVC$^{eGFP\Delta FHN+GFtail}$ at MOIs of 1.5, 3, and 6. Images were analyzed at 24 hpi. The nucleus was stained with Hoechst 33342 (blue), and green fluorescence indicates viral infection, as shown by eGFP expression. rSeVC$^{eGFP}$ and Mock-infected cells were used as controls. Scale bar lengths are indicated. (TIF)

**S1 Table. CRISPR screening data (nsVG negative stock).** (XLS)

**S2 Table. CRISPR screening data (nsVG negative stock + Ruxolitinib).** (XLS)

**S3 Table. CRISPR screening data (nsVG positive stock).** (XLS)

**S4 Table. Data sheet for graphs.** (XLSX)

## Acknowledgments

We thank Drs. Karl-Klaus Conzelmann for providing the BSR-T7/5 cells, Sean Whelan for the rVSV$^{eGFP}$ virus, Susan Weiss for the rNDV$^{eGFP}$ virus, and Benhur Lee for the rSeV-M-HA virus. Flow cytometry work was supported by Dr. Asya Smirnov and the cWIDR and Department of Molecular Microbiology Flow Cytometry Facility at Washington University School of Medicine.

## Author Contributions

**Conceptualization:** Carolina B. López.

**Data curation:** Carolina B. López.

**Formal analysis:** Yanling Yang, Leran Wang.

**Funding acquisition:** Carolina B. López.

**Investigation:** Yanling Yang, Yuchen Wang, Wandy Beatty.

**Methodology:** Yanling Yang, Danielle E. Campbell, Megan Baldridge.

**Project administration:** Carolina B. López.

**Resources:** Heng-Wei Lee, Megan Baldridge.

**Supervision:** Carolina B. López.

**Validation:** Yanling Yang.

**Writing – original draft:** Yanling Yang.

**Writing – review & editing:** Yanling Yang, Yuchen Wang, Heng-Wei Lee, Wandy Beatty, Megan Baldridge, Carolina B. López.

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
