## [Decision Letter · Decision Letter 0]

15 Oct 2024

Dear Dr. López,

Thank you very much for submitting your manuscript "SLC35A2 modulates paramyxovirus fusion events during infection" for consideration at PLOS Pathogens. As with all papers reviewed by the journal, your manuscript was reviewed by members of the editorial board and by several independent reviewers. In light of the reviews (below this email), we would like to invite the resubmission of a significantly-revised version that takes into account the reviewers' comments.

The reviewers agreed that the manuscript reports very interesting findings that will spur further research in the area. Each reviewer, however, expressed concerns regarding the clarity of data presentation and conclusions. These would need to be addressed. The reviewer 1, in particular, drew attention to the convoluted experimental approach that made it difficult to compare outcomes across different viruses and assign them to a particular experimental variable. A report that is confusing to experts in the field of paramyxovirus entry will be even more difficult to parse for a broad audience. Therefore, the authors are encouraged to carefully reword confusing points and provide additional experimental support wherever needed to reach clearer conclusions.

We cannot make any decision about publication until we have seen the revised manuscript and your response to the reviewers' comments. Your revised manuscript is also likely to be sent to reviewers for further evaluation.

Sincerely,

Ekaterina E. Heldwein

Academic Editor

PLOS Pathogens

Matthias Schnell

Section Editor

PLOS Pathogens

Michael Malim

Editor-in-Chief

PLOS Pathogens

orcid.org/0000-0002-7699-2064

The reviewers agreed that the manuscript reports very interesting findings that will spur further research in the area. Each reviewer, however, expressed concerns regarding the clarity of data presentation and conclusions. These would need to be addressed. The reviewer 1, in particular, drew attention to the convoluted experimental approach that made it difficult to compare outcomes across different viruses and assign them to a particular experimental variable. A report that is confusing to experts in the field of paramyxovirus entry will be even more difficult to parse for a broad audience. Therefore, the authors are encouraged to carefully reword confusing points and provide additional experimental support wherever needed to reach clearer conclusions.

Reviewer's Responses to Questions

**Part I - Summary**

Reviewer #1: The paper by Yang et al makes a nice contribution to understanding host molecules that affect paramyxovirus entry/fusion.

The set of experiments that is described here first detail a loss of function screen for host factors in paramyxovirus infection, and identify two transporters involved in adding carbohydrates (ultimately, sialic acids) to glycoproteins and glycolipids, encoded by SLC35A1 and SLC35A2. Both are essential for infection by sendai virus (a model paramyxovirus). Host cell sialic acid processing (and expression on cell surfaces) has been known to be critical for paramyxovirus entry, and these findings make sense in the context of what is known. The paper then proceeds to assess the distinction between the roles of these two transporters, and the effects of KO of these transporters, on a series of different paramyxoviruses (NDV and mumps in addition to Sendai). The findings are interesting in that they extend the notion of the differences in receptor density requirements in different infection conditions and between viruses; the differences in receptor density requirements for viral entry vs. viral spread; and the comparison of these properties among sialic acid-binding paramyxoviruses.

The experiments raise interesting questions that will merit further investigation. A number of mechanistic questions remain open. I have a few questions below that the authors may wish to address.

1. There is an intersecting set of factors that affect the outcome of infection in this paper, and the roles of (a) cell surface sialylated receptor type; (b) cell surface receptor density; (c) cell type are entangled as to cause/effect. I would like this to be somewhat clearer.

2. The confirmation that sialic acid moieties on the surface are essential host factors may be more firmly placed in context of earlier discoveries. This portion of the paper could be shortened significantly so that the paper can focus on the more novel parts.

3. The impact of the KO experiments directly depends on the specific cell lines used in each experiment, as entry/fusion is highly dependent on the specific cell and composition of the cell membrane. The systems used here are highly lab adapted – and while it’s totally justifiable to use model systems, the comparisons are valid only in relative terms and if the comparisons are made of infections in the same cell lines. This should be clarified so that it’s evident which factors are being held constant.

4. Just as the infection is affected by the KO of one or the other sia transporter, the cell line being used and the virus strain being used also impacts receptor usage and the ratio of entry to spread (etc) and these factors (cell line, virus strain) should be carefully discussed.

5. In terms of receptor usage (e.g.lines 196-9) it seems that the authors are implying but not directly saying that there is a receptor usage difference between the paramyxoviruses that are tested. This should be clearly stated with the references to receptors differentially used and their sia status (if known) to make sense of these findings.

6. The sections on differential infection vs spread (e.g.lines 213-222) are very interesting and merit development. There have been various thoughts for paramyxoviruses about the different requirements for entry vs. spread, and publications that relate to this question. As above, this will differ greatly (entry vs. spread for each of these viruses) depending on the cell line used (and the specific strain). This topic in the paper needs clarity and would benefit from some additional development and discussion.

7. Regarding this topic for mumps infection (e.g. lines 290-299) it would be of interest to know here, what do the differential responses to entry/infection vs. syncytia production say about receptor usage by these viruses? And, are the syncytia biologically meaningful or are they are tissue culture artifact – or a surrogate (as the authors imply) for a persistent mode of infection? These questions should be clarified.

8. The work contains many fascinating observations and offers interesting observations about these processes for several lab strain paramyxoviruses in cell lines. It would benefit the paper if the conclusions that may be drawn and the lines of evidence could be cleanly summarized and the points above about the influence of cell line vs viral receptor usage could be integrated into those conclusions. Currently the information is somewhat diffusely presented and some of the potentially very interesting aspects of these comparisons are not summarized. Regardless of the artifactual nature of the systems, if the viruses / pathways are placed on a level field, the comparisons are potentially very interesting.

Reviewer #2: The authors have used a loss-of-function CRSPR screen to identify host factors critical for paramyxovirus infection. They used Sendai virus for the initial screen in A549 cells, where they identified CMP-sialic acid transporter (CST) gene SLC35A1 and the UDP-galactose transporter (UGT) gene SLC35A2 as essential for Sendai virus infection. They then proceeded to validate these hits in mumps virus and Newcastle disease virus. They made single and double knockout cells and proceeded to show that SLC35A1 was essential for virus binding, as knocking it out eliminated cell surface sialic acid, the receptor for the viruses tested. The importance of SLC35A1 in influenza virus and porcine delta coronavirus binding and entry have already been reported. On the other hand, SLC35A2 seemed to be involved in virus-cell fusion in the case of Sendai virus and cell-cell fusion in the case of mumps virus. This is the first report on the possible role of SLC35A2 in membrane fusion and as such is an intriguing finding that warrants further study.

Reviewer #3: Yang et al. found that sugar nucleotide transporters, SLC35A1 and SLC35A2, were critical factors facilitating various paramyxoviruses through the genome-wide CRISPR KO. The authors mentioned that SLC35A2, the UDP-galactose transporter, played some roles in the viral fusion step. The observation was potentially interesting; however, some of the descriptions were not fully supported by the experimental results.

**Part II – Major Issues: Key Experiments Required for Acceptance**

Reviewer #1: (No Response)

Reviewer #2: Line 243 and Fig 6A: The authors state that SeV binding to SLC35A2 KO cells was lower than that of control cells. It may be helpful to the reader if this was quantified, perhaps as mean fluorescence intensity (MFI).

Reviewer #3: Through the manuscript, the effect of SLC35A2 on the fusion activity of SeV was evidenced via the results of Fig. 3B. However, the settings of 400 MOI were unlikely in routine experiments. Since this experiment provided very important evidence for the core statement of this manuscript, the result should be confirmed by multiple experimental methods.

The author evaluated cellular glycans via staining with lectins. However, KO of a certain glycan-synthesis machinery often resulted in unexpected changes in global cellular glycome. I highly encourage the authors to evaluate the cellular glycans using glycomic approaches such as MS or lectin array.

I could not find a detailed description of miRF670.

This is a optional comment. To clarify the role of SLC35A2 on viral infectivity, propagating the virions in SLC35A2-KO cells and checking the infectivity of the virions in other permissive cells perhaps make sense. If the glycosylations of the virions critically contribute to the loss of function, the fusion process would be disabled in the virions propagated in the KO cells. In the other case, cellular SLC35A2 would support the fusion process in some way.

**Part III – Minor Issues: Editorial and Data Presentation Modifications**

Reviewer #1: (No Response)

Reviewer #2: 1. Line 56: Italicize SLC35A1.

2. Line 71-74: While the authors state that they used a rSeV that allowed for more sensitive and accurate analysis of CRISPR-Cas9 knockout, it is not clear how this is achieved using this construct. Part of the answer is revealed in the Discussion (lines 368-370). The authors should consider moving this explanation to the Introduction section where the rSeV is first described or to the Results section when the authors describe the use if this rSeV in the CRSIPR screen.

3. Fig 1B and 1D are not called out in the main text.

4. Line 164: rSeVCmiRF670 was abruptly introduced without an explanation of what this construct is and why it was used.

5. Line 190: “look” should be “looked”.

6. Line 194-196: This statement is difficult to follow. Please rephrase to help the reader.

7. Line 257: “affects” should be “affect”.

8. Line 267-8 and Line 319: These statements should be qualified to indicate that SLC35A2 does increase the efficiency of SeV binding to cells.

9. Line 313: “transporters” should be “transporter”.

10. Line 326: delete “a”.

11. The authors use both “dsGFP” and “dseGFP” in the text. Are these the same construct or different?

12. Discussion of how SLC35A2 may impact membrane fusion was lacking.

Reviewer #3: The experiment in Fig. 3B is hard to understand. An illustration of the principle of the experiment will help the readers easily understand.

PLOS authors have the option to publish the peer review history of their article (what does this mean?). If published, this will include your full peer review and any attached files.

Reviewer #1: No

Reviewer #2: No

Reviewer #3: No
---

## [Editor Report · Decision Letter 1]

25 Dec 2024

Dear Dr. López,

We are pleased to inform you that your manuscript 'SLC35A2 modulates paramyxovirus fusion events during infection' has been provisionally accepted for publication in PLOS Pathogens.

Best regards,

Ekaterina E. Heldwein

Academic Editor

PLOS Pathogens

Matthias Schnell

Section Editor

PLOS Pathogens

Sumita Bhaduri-McIntosh

Editor-in-Chief

PLOS Pathogens

orcid.org/0000-0003-2946-9497

Michael Malim

Editor-in-Chief

PLOS Pathogens

orcid.org/0000-0002-7699-2064
---

## [Editor Report · Acceptance letter]

2 Jan 2025

Dear Dr. López,

We are delighted to inform you that your manuscript, "SLC35A2 modulates paramyxovirus fusion events during infection," has been formally accepted for publication in PLOS Pathogens.

Best regards,

Sumita Bhaduri-McIntosh

Editor-in-Chief

PLOS Pathogens

orcid.org/0000-0003-2946-9497

Michael Malim

Editor-in-Chief

PLOS Pathogens

orcid.org/0000-0002-7699-2064